# Suppression of *C9orf72* RNA repeat-induced neurotoxicity by the ALS-associated RNA-binding protein Zfp106

Barbara Celona[1], John von Dollen[2], Sarat C Vatsavayai[3], Risa Kashima[1], Jeffrey R Johnson[2], Amy A Tang[4], Akiko Hata[1], Bruce L Miller[3], Eric J Huang[4], Nevan J Krogan[2], William W Seeley[3,4], Brian L Black[1,5]*

[1]Cardiovascular Research Institute, University of California, San Francisco, San Francisco, United States; [2]Department of Cellular and Molecular Pharmacology, University of California, San Francisco, San Francisco, United States; [3]Department of Neurology, University of California, San Francisco, San Francisco, United States; [4]Department of Pathology, University of California, San Francisco, San Francisco, United States; [5]Department of Biochemistry and Biophysics, University of California, San Francisco, San Francisco, United States

*For correspondence: brian.black@ucsf.edu

Competing interests: The authors declare that no competing interests exist.

**Abstract** Expanded GGGGCC repeats in the first intron of the *C9orf72* gene represent the most common cause of familial amyotrophic lateral sclerosis (ALS), but the mechanisms underlying repeat-induced disease remain incompletely resolved. One proposed gain-of-function mechanism is that repeat-containing RNA forms aggregates that sequester RNA binding proteins, leading to altered RNA metabolism in motor neurons. Here, we identify the zinc finger protein Zfp106 as a specific GGGGCC RNA repeat-binding protein, and using affinity purification-mass spectrometry, we show that Zfp106 interacts with multiple other RNA binding proteins, including the ALS-associated factors TDP-43 and FUS. We also show that *Zfp106* knockout mice develop severe motor neuron degeneration, which can be suppressed by transgenic restoration of Zfp106 specifically in motor neurons. Finally, we show that Zfp106 potently suppresses neurotoxicity in a *Drosophila* model of *C9orf72* ALS. Thus, these studies identify Zfp106 as an RNA binding protein with important implications for ALS.

## Introduction

Expanded GGGGCC repeats in the first intron of the *C9orf72* gene represent the most common cause of familial amyotrophic lateral sclerosis (ALS) and frontotemporal dementia (FTD) (*DeJesus-Hernandez et al., 2011*; *Renton et al., 2011*). The repeat expansion has been identified in more than 40% of familial ALS and in at least 7% of sporadic cases (*Majounie et al., 2012*). Several gain-of-function mechanisms for how the expanded hexanucleotide repeats cause disease have been proposed. One idea suggests that toxic dipeptide repeat (DPR) proteins generated by repeat associated non-ATG (RAN) translation of the RNA repeats cause neurodegeneration (*Kwon et al., 2014*; *Mizielinska et al., 2014*; *Mori et al., 2013*). Another major proposed gain-of-function mechanism for *C9orf72* ALS is via the formation of pathogenic repeat RNA aggregates that sequester one or more RNA binding proteins, leading to altered RNA processing and metabolism in motor neurons (*Cooper-Knock et al., 2015*; *Haeusler et al., 2014*; *Lee et al., 2013*; *Prudencio et al., 2015*; *Sareen et al., 2013*).

Zfp106 is a $C_2H_2$ zinc finger protein with four predicted zinc fingers and seven WD40 domains (*Grasberger and Bell, 2005*). The mouse *Zfp106* gene is located on chromosome 2 at 60.37 cM

**eLife digest** Molecules of ribonucleic acid (or RNA for short) have many roles in cells, including acting as templates to make proteins. RNA is made of building blocks called nucleotides that are assembled to form strands. The precise order of the nucleotides in an RNA molecule can have a dramatic effect on the role that RNA plays in the body. For example, amyotrophic lateral sclerosis (ALS) is a deadly disease caused by the gradual loss of the nerve cells that control muscle (known as motor neurons). The most common cause of inherited ALS is a genetic mutation that results in some RNA molecules having many more copies of a simple six nucleotide sequence known as GGGGCC than normal cells. RNA molecules with these "GGGGCC repeats" form clumps in motor neurons.

The clumps of RNA molecules also contain proteins, but the identities of these RNA-binding proteins and the roles they play in ALS remain largely unknown. Celona et al. have now identified a new RNA-binding protein called Zfp106, which binds specifically to GGGGCC repeats in mice and fruit flies.

Removing the gene that encodes Zfp106 from mice causes the mice to develop ALS. On the other hand, restoring Zfp106 only to the motor neurons of these mutant mice prevents the mice from developing disease. This suggests that Zfp106's role is specific to motor neurons. Indeed, fruit flies that have too many copies of GGGGCC develop severe symptoms reminiscent of ALS. Introducing a mammalian version of Zfp106 into these flies prevents them from developing the disease.

The findings of Celona et al. suggest that Zfp106 might be a potential new drug target for treating ALS in humans. The next step following this work will be to find out exactly how Zfp106 regulates normal cellular processes by binding to RNA and how it suppresses ALS-like disease by binding to GGGGCC RNA-repeats.

(Chr2:120,506,820–120,563,843 bp; Genome Reference Consortium, Mouse Build 38). In addition, the human ortholog, *ZNF106*, is located at human chromosome 15q15.1 (Chr15: 42,412,437–42,491,197; Genome Reference Consortium Human Build 38), a region with strong linkage to a rare recessive familial form of ALS (*Hentati et al., 1998*), suggesting a possible role in human ALS. Consistent with this notion, mice lacking *Zfp106* function show evidence of nondevelopmental neuro-muscular degeneration, also suggestive of a possible role for Zfp106 in ALS (*Anderson et al., 2016*; *Joyce et al., 2016*; *van der Weyden et al., 2011*).

Here, we identify a previously unknown role for Zfp106 as an RNA binding protein that specifically interacts with GGGGCC repeats and with a network of ALS-associated RNA-binding proteins, including TDP-43. In addition, we establish that knockout of *Zfp106* in mice results in an ALS-like motor phenotype and that transgenic restoration of Zfp106 expression specifically in motor neurons suppresses the neurodegenerative phenotype in knockout mice. Finally, we show that Zfp106 is a potent suppressor of neurotoxicity caused by expression of GGGGCC repeats in a *Drosophila* model of *C9orf72* ALS. Thus, these studies have important implications for the most common form of human ALS.

## Results and discussion

Sequestration of RNA binding proteins by rGGGGCC repeats has been implicated in the pathology of *C9orf72* ALS (*Mizielinska and Isaacs, 2014*). Therefore, to identify rGGGGCC-binding proteins that may be sequestered by repeat-containing RNA and involved in the pathology of *C9orf72* ALS, we performed a pull-down assay using biotinylated $r(GGGGCC)_8$ RNA and identified the zinc finger protein Zfp106 as a previously unknown rGGGGCC-binding protein. Zfp106 bound specifically to $r(GGGGCC)_8$ but not the control sequence $r(AAAACC)_8$ (*Figure 1a*). RNA electrophoretic mobility shift assay (EMSA) using purified Zfp106 protein demonstrated that Zfp106 bound directly to the rGGGGCC repeats and that this binding was specific since it was efficiently competed by unlabeled self oligonucleotide but not a $30\times$ molar excess of the control oligonucleotide (*Figure 1b*). Importantly, Zfp106 binding to $r(GGGGCC)_4$ was not competed by $30\times$ molar excess of a mutated RNA probe of equal length, nor was it competed by $30\times$ molar excess of

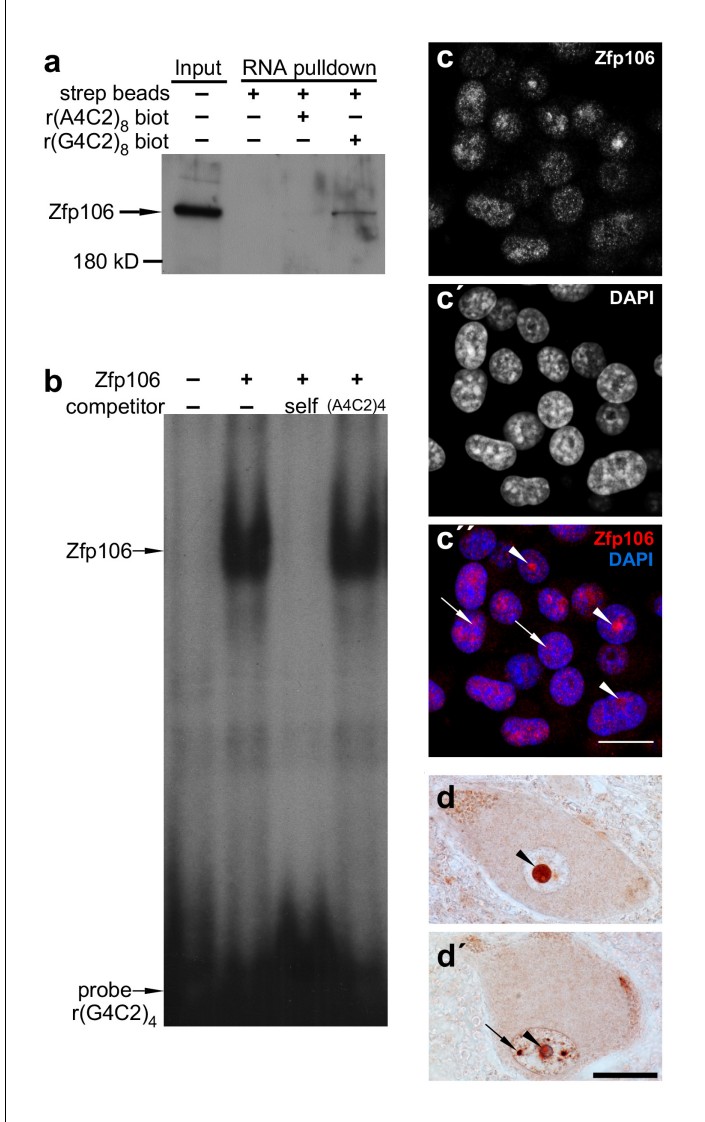

**Figure 1.** Zfp106 is an rGGGGCC binding protein. (**a**) Western blot analysis of Zfp106 eluted from a biotinylated-RNA pulldown of Neuro-2a nuclear proteins. Endogenous Zfp106 was specifically pulled down by $(GGGGCC)_8$ biotinylated RNA and not by the unrelated $(AAAACC)_8$ biotinylated RNA oligonucleotide. (**b**) RNA EMSA performed with purified Zfp106 protein demonstrates that Zfp106 directly binds $(GGGGCC)_4$ RNA in vitro. $30\times$ molar excess of unlabeled self competitor, but not $30\times$ $r(AAAACC)_4$, competes with Zfp106 binding to $r(GGGGCC)_4$, establishing specificity of the interaction. (**c**) Immunofluorescence with an anti-Zfp106 antibody detects Zfp106 expression in the nucleolus (arrowheads) and in other discrete nuclear puncta (arrows) in cultured Neuro-2a cells. The three images are the same section showing the red channel in (**c**), the blue channel in (**c′**), and both channels in (**c″**). Scale bar, 20 μm. (**d, d′**) Immunohistochemical detection of human ZNF106 shows strong nucleolar expression in human motor neurons in the anterior horn of spinal cord. The two images are from different post-mortem individuals. Arrowheads mark nucleolar ZNF106 expression; arrows mark other nuclear and perinucleolar foci of ZNF106 expression. Scale bar, 25 μm.

The following figure supplement is available for figure 1:

**Figure supplement 1.** Zfp106 binds specifically to rGGGGCC repeats.

rCUG repeats comprising a MBNL1 binding site (*Figure 1—figure supplement 1a*). Likewise, MBNL1, an unrelated RNA-binding protein involved in myotonic dystrophy (*Miller et al., 2000*), bound efficiently and specifically to rCUG repeats under conditions in which Zfp106 showed no binding to the same rCUG sequence (*Figure 1—figure supplement 1b*) and, as previously described (*Reddy et al., 2013*), was not able to bind GGGGCC repeats (*Figure 1—figure supplement 1a*). Together, these data demonstrate that Zfp106 binds efficiently and specifically to GGGGCC RNA repeats.

Patients with *C9orf72* ALS show evidence of post-mortem nucleolar stress (*Haeusler et al., 2014*). Interestingly, in cultured Neuro-2a neuronal cells, Zfp106 was primarily present in the nucleolus and in other discrete nuclear puncta (*Figure 1c*). Likewise, in lower motor neurons from post-mortem human spinal cord samples, ZNF106 was primarily observed in the nucleolus (*Figure 1d,d'*) and also showed occasional localization in other puncta in the nucleus outside of the nucleolus (*Figure 1d'*). The localization of Zfp106 to the nucleolus is in accordance with previous studies showing that Zfp106 is predominantly located in the nucleolus of C2C12 myoblast cells (*Grasberger and Bell, 2005*), and taken together, the results presented in *Figure 1* support a possible role for Zfp106 in the biology of *C9orf72* neurodegeneration.

As noted above, previous studies have demonstrated a requirement for *Zfp106* in neuromuscular function in mice. Abnormal skeletal morphology in *Zfp106*$^{tm1a(KOMP)Wtsi}$ homozygous mice was briefly reported as part of a large-scale phenotypic screen of genetically modified mice by the Sanger Mouse Genetics Project (*van der Weyden et al., 2011*). More recently, more extensive phenotypic studies demonstrated progressive neuromuscular disease in the absence of *Zfp106* gene function (*Anderson et al., 2016*; *Joyce et al., 2016*). These studies, combined with the interaction of Zfp106 with the *C9orf72* hexanucleotide repeat sequence, prompted us to examine *Zfp106* knockout mice in additional detail and to determine whether the observed defects were autonomous to motor neurons. Therefore, we obtained the previously described *Zfp106*$^{tm1a(KOMP)Wtsi}$ mice from the KOMP knockout consortium (*van der Weyden et al., 2011*) and generated the presumptive null allele (*Zfp106*$^{lacz}$; also referred to herein as *Zfp106*-null or *Zfp106*$^{-/-}$) using a universal germline deleter transgenic mouse line (*Ehlers et al., 2014*).

Consistent with published studies, we found that loss of *Zfp106* function caused a progressive neuromuscular phenotype. *Zfp106*-null mice were born at normal Mendelian frequency and were indistinguishable from control mice until approximately 4 weeks of age, when 100% of knockout mice began to exhibit evidence of a wasting disease and loss of muscle strength (*Figure 2—figure supplement 1*; *Video 1*). Transverse sections of the quadriceps from 4-week-old *Zfp106*-null mice showed increased variation in fiber size and evidence of grouped atrophy with very few centrally located nuclei (*Figure 2a,b*), consistent with neurogenic rather than myopathic disease (*Baloh et al., 2007*). By three months of age, near end stage disease, *Zfp106*-null mice displayed extensive muscular atrophy and severe degeneration of muscle fibers (*Figure 2c,d*). The number of choline acetyltransferase (ChAT)-positive motor neurons was also profoundly and significantly reduced in *Zfp106* knockout spinal cords compared to littermate control mice (*Figure 2e–i*), providing further evidence that loss of Zfp106 in mice causes a neurodegenerative phenotype. Independently, we confirmed the knockout phenotype observed with the KOMP allele (*van der Weyden et al., 2011*) using CRISPR-mediated gene disruption by non-homologous end joining (NHEJ) and creation of a frameshift mutation leading to a premature stop codon. The CRISPR-generated mutation resulted in an essentially identical wasting and histological phenotype (data not shown).

To gain insight into the cell autonomous requirements for Zfp106, we used the KOMP

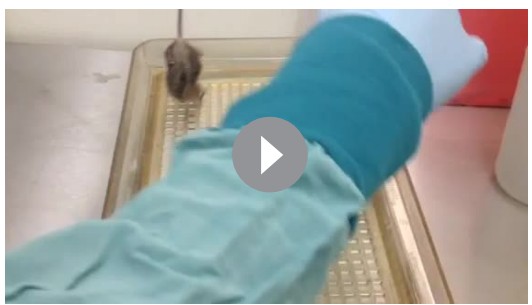

**Video 1.** *Zfp106*-null mice exhibit a severe progressive neurodegenerative phenotype. The video shows a Zfp106 knockout mouse and a littermate control at 10 weeks of age. The Zfp106 knockout displays abnormal gait due to hind limb paralysis, severe kyphosis, wasting, and persistent hind limb clasping at tail suspension. The wild type mouse displays normal gait and normal extension of the hind limbs away from the abdomen during the tail suspension test.

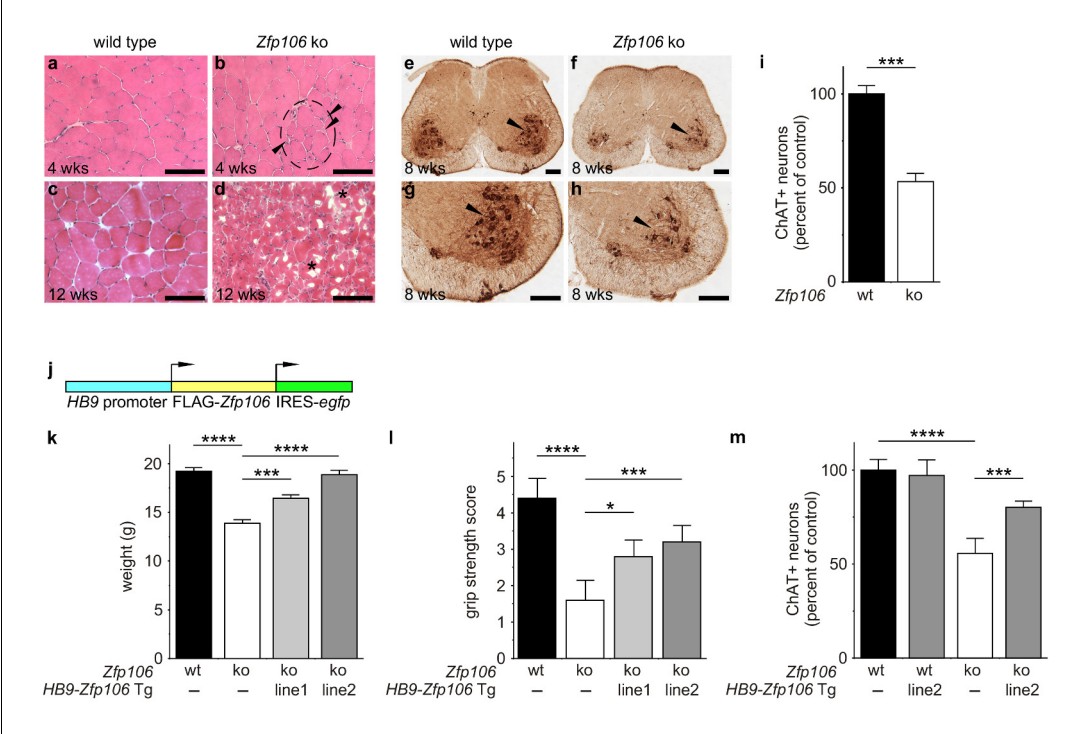

**Figure 2.** Zfp106 loss-of-function causes a severe neuromuscular phenotype in mice. (a–h) Evidence of profound neuromuscular pathology in *Zfp106*-null mice. H&E stained sections of quadriceps muscles (a–d) from 4-week old and 12-week old wild type (a, c) and *Zfp106* knockout (b, d) mice are shown. Note the presence of grouped small angular fibers in knockout mice (dashed circle), already clearly evident by 4-weeks of age (b, arrowheads) compared to the rounder, more evenly-shaped fibers in controls (a). By 12-weeks (end stage disease), *Zfp106* knockout mice have profound pathology and the presence of atrophic and degenerated muscle fibers with numerous inclusions (d, asterisks). Scale bar, 100 μm. ChAT immunohistochemical staining (e–h) of lumbar spinal cord sections from 8-week old wild type (e, g) and Zfp106 knockout (f, h) mice show a drastic reduction in the number of ChAT+ motor neurons. Arrowheads mark ChAT-expressing motor neurons in the ventral horn. Scale bar, 200 μm in all panels. (i) Quantification of ChAT-positive neurons in the lumbar spinal cord of wild type and *Zfp106* knockout mice at 8 weeks of age shows a ~ 47% loss of motor neurons in *Zfp106* knockout mice. *n* = 5 mice for each genotype. Data are expressed as percent of control ± SD and were analyzed by *t*-test. ***p<0.001. (j) Schematic of the *HB9-3×FLAG-2×STREP-Zfp106-IRES-gfp* (*HB9-Zfp106*) transgene. (k, l, m) Transgenic expression of Zfp106 in motor neurons suppresses the wasting phenotype (k), the reduction in grip strength (l) and the loss of lumbar spinal cord motor neurons (m) in *Zfp106* knockout mice. Wasting data are expressed as the mean weight ± SEM. Grip strength data are expressed as the mean grip score ± SD; grip score: 1, 1–10 s; 2, 11–25 s; 3, 26–60 s; 4, 61–90 s; 5, > 90 s. ChAT-positive neuron counts are expressed as percent of control ± SD. *n* = 5 female mice at 8 weeks of age for each genotype. Data were analyzed by one-way ANOVA and Bonferroni's Multiple Comparison Test. *p<0.05; ***p<0.001, ****p<0.0001.

The following figure supplements are available for figure 2:

**Figure supplement 1.** *Zfp106* knockout mice exhibit progressive and severe weight and grip strength loss over time.

**Figure supplement 2.** *Zfp106* is expressed in skeletal muscle and motor neurons.

**Figure supplement 3.** Expression of Zfp106 in wild type, knockout and *HB9-Zfp106* transgenic mice.

**Figure supplement 4.** Rescue of ChAT+ motor neuron loss by restoration of Zfp106 to motor neurons in *Zfp106* knockout mice.

*lacZ* knock-in allele (*van der Weyden et al., 2011*) to examine *Zfp106* expression using β-galactosidase activity as a sensitive method for detecting expression. This approach showed *Zfp106* expression in both skeletal muscle and motor neurons (*Figure 2—figure supplement 2a,b*). Given the multiple lines of evidence for a role of Zfp106 in neurodegeneration, and the evident denervation observed in the knockout mice, we hypothesized that the primary requirement for Zfp106 was likely to be in motor neurons. We confirmed a highly significant and dramatic reduction in *Zfp106* expression in spinal cord (*Figure 2—figure supplement 3a*). The ~80% reduction in *Zfp106* expression is

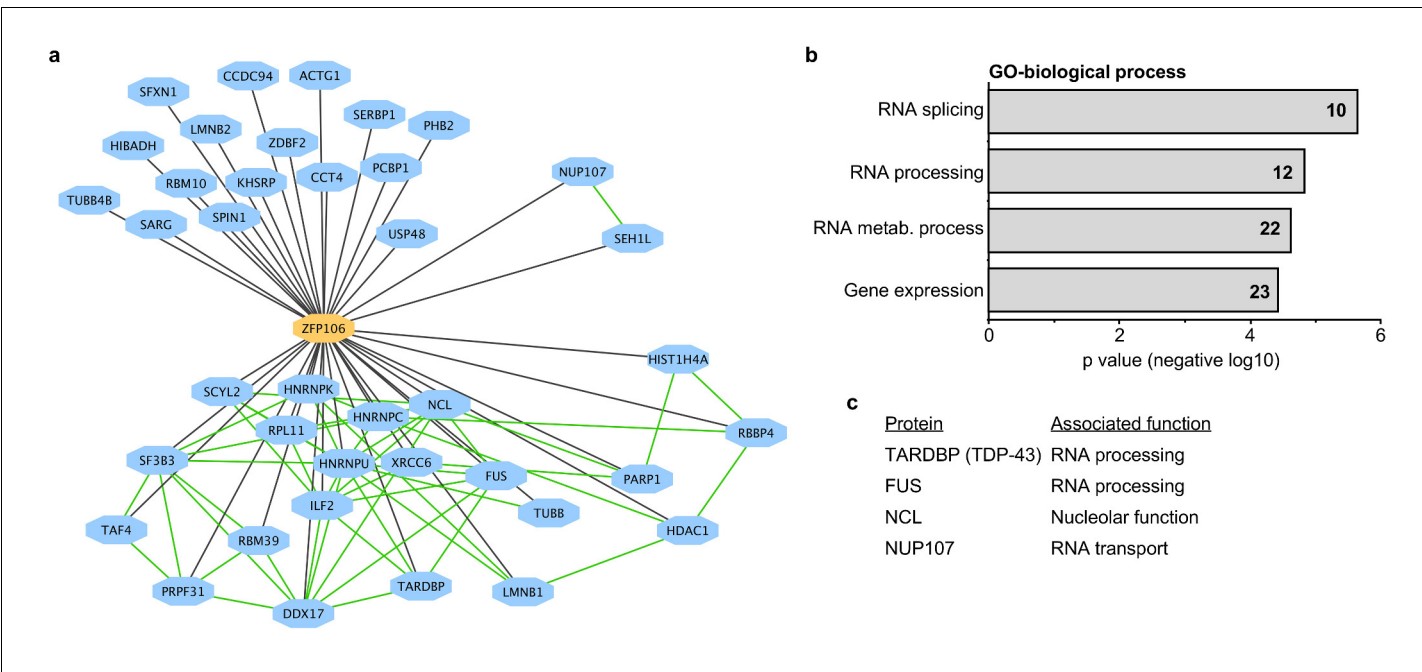

**Figure 3.** Zfp106 interacts with a network of other ALS-associated RNA-binding proteins. (**a**) Network representation of the Zfp106 interactome generated by Cytoscape. Zfp106 (yellow node) interacts with 39 proteins with top 5% MiST score (blue nodes). Curated protein-protein interactions from the CORUM database are indicated as nodes connected by green lines. (**b**) Gene Ontology (GO) Biological Process analysis of the Zfp106 interactome ranked by p value. The number of interacting proteins (top 5% by MiST score) in each category is indicated inside each bar. (**c**) Selected ALS-associated proteins within the top 5% MiST-scoring hits from the Zfp106 interactome are listed with putative functions associated with the pathobiology of ALS indicated.

The following figure supplement is available for figure 3:

**Figure supplement 1.** Zfp106 interacts with TDP43 and Nup107 in Neuro-2a cells.

consistent with previous studies (*Anderson et al., 2016*; *Joyce et al., 2016*). Therefore, we used a transgenic approach in mice to restore Zfp106 expression only in motor neurons under the control of the *Mnx1* promoter (herein called *HB9*, *HB9-Zfp106*) on a *Zfp106* germline knockout background (*Figure 2j*; and *Figure 2—figure supplement 3b,c*). Importantly, restoration of Zfp106 specifically in motor neurons significantly suppressed the *Zfp106* knockout wasting and muscle grip phenotypes and resulted in restoration of the number of ChAT-expressing motor neurons (*Figure 2j–m*; *Figure 2—figure supplement 4*), indicating a cell autonomous requirement for Zfp106 in motor neurons.

To gain additional insights into the molecular function of Zfp106, we used mass spectrometry to identify all proteins co-immunoprecipitated by Zfp106 from 293T cells (*Figure 3*). Several controls were included in the mass spectrometry experiments, including the DNA-binding factors Etv2 and MEF2C and the RNA-binding proteins Vif and Tat. The interactomes for each of the controls were compared to the Zfp106 interactome using the previously described MiST algorithm, which determines specific interactions based on enrichment, reproducibility, and specificity compared to interaction with the negative controls (*Jager et al., 2012*). These experiments identified thirty-nine proteins that specifically interacted with Zfp106 (*Figure 3a*; *Supplementary file 1*). Gene ontology (GO) and protein complex analyses revealed a significant enrichment in interactors with roles in RNA processing, metabolism, and splicing (*Figure 3a,b*; *Supplementary file 2*), suggesting that Zfp106 may also play roles in these processes.

There is growing and compelling genetic and pathological evidence that dysregulated RNA processing and metabolism contribute to the development of ALS and *C9orf72* ALS in particular (*Ling et al., 2013*; *Prudencio et al., 2015*). Interestingly, several of the Zfp106-interacting proteins identified in our affinity purification-mass spectrometry experiment are proteins directly associated

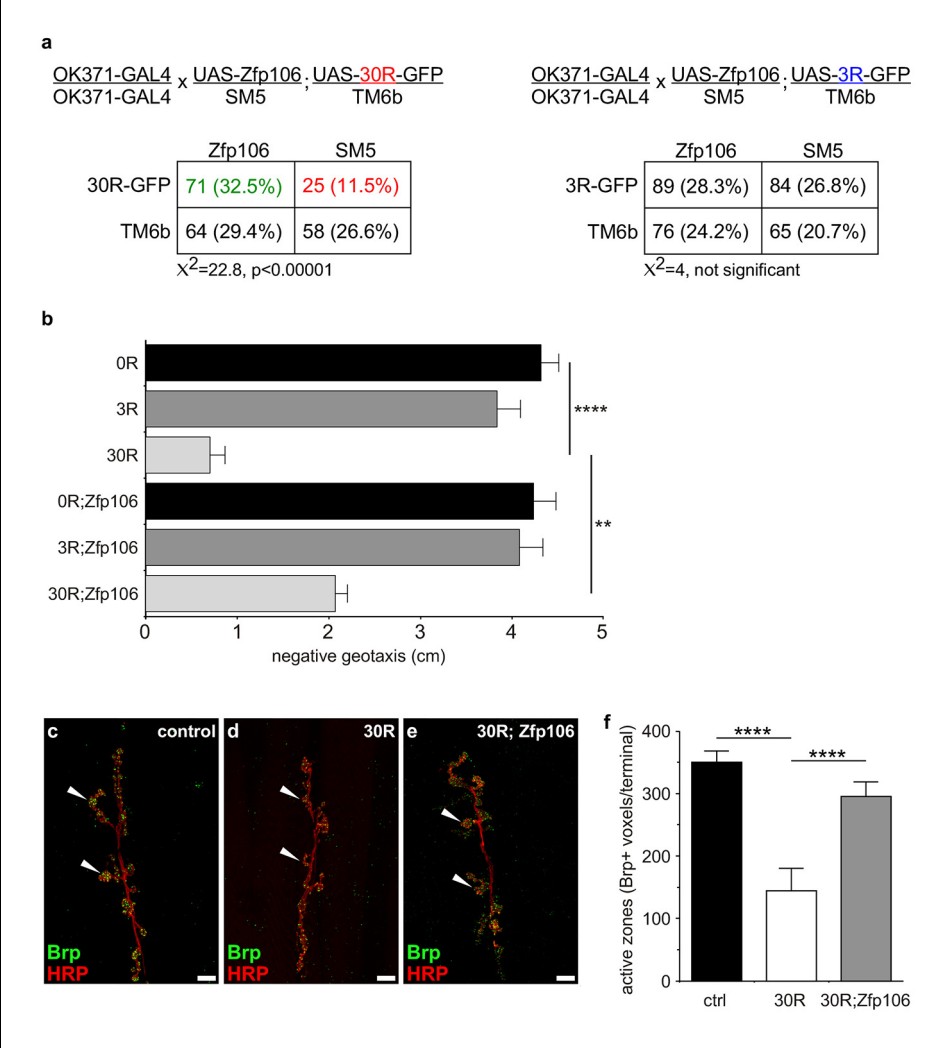

**Figure 4.** Zfp106 suppresses GGGGCC-mediated neurotoxicity in Drosophila. (**a**) Schematics and Punnett squares from crosses of UAS-Zfp106; UAS-30R-GFP (left) and UAS-Zfp106; UAS-3R-GFP (right) transgenic flies to OK371-GAL4 flies. The Punnett squares show the number of flies that eclosed from the indicated crosses and show that expression of 30R-GFP resulted in failure of ~50% of flies to eclose (25 actual versus 54.5 predicted); this defect was completely suppressed by co-expression of Zfp106. The Chi-square test ($\chi^2$) was used to determine the significance of the differences between the observed and expected frequencies of the expected genotypes. (**b**) Locomotor activity of 28-day post-eclosion flies of the indicated genotypes assessed by rapid iterative negative geotaxis (RING) assay. Data are presented as the mean height climbed ± SEM. A two-way ANOVA analysis of the locomotor data revealed a significant effect of repeat expression (p<0.0001) and Zfp106 expression (p<0.01) on locomotor activity, with a significant interaction between the two factors (p<0.01); i.e. Zfp106 coexpression significantly suppresses neurotoxicity induced by expression of 30R (p<0.01) by Bonferroni's posthoc Multiple Comparison Test. n = 25 male flies for each genotype; ****p<0.0001, **p<0.01. (**c, d, e**) Representative NMJs of muscle 6/7 in abdominal segment A3 of third instar larvae of the indicated genotypes co-stained with anti-HRP (neuronal marker, red channel) and anti-Bruchpilot (Brp) (active zone marker, green channel). Staining of the active zone component Brp shows that expression of 30R causes a reduction in Brp-positive active zones [arrowheads in (**d**) compared to control in **c**]. Coexpression of Zfp106 suppressed the reduction in Brp-positive active zones caused by 30R [arrowheads in **e** compared to **d**]. (**f**) Quantification of active zones as total voxel occupancy by Brp signal at each terminal shows that Zfp106 significantly suppresses the reduction in active zone caused by expression of 30R. Data were analyzed by one-way ANOVA and Bonferroni's Multiple Comparison Test. ***p<0.001, ****p<0.0001.

The following figure supplement is available for figure 4:

**Figure supplement 1.** GAL4-dependent expression of Zfp106 in *Drosophila* does not affect 30R-*gfp* mRNA levels.

with ALS and/or *C9orf72* ALS, including splicing factors TDP-43 and FUS, the nucleolar protein Nucleolin, and the nuclear pore component Nup107 (*Freibaum et al., 2015*; *Haeusler et al., 2014*; *Lagier-Tourenne et al., 2010*; *Ling et al., 2013*); *Figure 3c*; *Supplementary file 1*). We further examined the interaction of Zfp106 with TDP-43 and Nup107, as examples of interacting proteins, by anti-Flag-Zfp106 immunoprecipitation followed by western blot for TDP43 and Nup107 in Neuro-2a cells (*Figure 3—figure supplement 1*). These experiments confirmed and validated the interaction with these other ALS-associated proteins and support the notion that Zfp106 is part of an RNA metabolic pathway (or pathways) that when dysregulated leads to neurodegeneration.

Prior studies have used Drosophila as an in vivo gain-of-function model of *C9orf72* neurodegeneration (*Freibaum et al., 2015*; *Tran et al., 2015*; *Xu et al., 2013*; *Zhang et al., 2015*). Here, we used the GAL4-UAS system to coexpress mouse Zfp106 and GGGGCC repeats in the fly in a gain-of-function approach to determine if the interaction between Zfp106 and GGGGCC repeats had a functional consequence in this model (*Figure 4*; *Figure 4—figure supplement 1a*). Expression of 30 tandem copies of the GGGGCC hexanucleotide in the 5′-untranslated region of a UAS-dependent *gfp* transcript (30R-GFP) in glutamatergic neurons using OK371-GAL4 (*Mahr and Aberle, 2006*) caused ~50% lethality due to an apparent failure of flies to eclose from the pupal case (*Figure 4a*). Furthermore, 28 day-old OK371; 30R-GFP-expressing flies that did manage to successfully eclose exhibited profound defects in locomotor activity compared to control flies (*Figure 4b*). No overt eclosion or locomotor phenotype was observed in flies expressing GFP only, 3R-GFP, or Zfp106 alone in glutamatergic neurons (*Figure 4a,b*). Previous studies have also shown that expression of 30R-GFP causes a reduction in the number of active zones in Drosophila larval neuromuscular junctions (*Zhang et al., 2015*). Consistent with those studies, we found that expression of 30R-GFP reduced active zones as measured by anti-Bruchpilot (Brp) immunofluorescence and quantification (*Figure 4c–f*). Remarkably, co-expression of Zfp106 with 30R-GFP restored Brp expression at larval active zones (*Figure 4c–f*) and completely suppressed the ~50% lethality caused by expression of 30R-GFP (*Figure 4a*). Moreover, co-expression of Zfp106 partially suppressed the adult locomotor defect caused by expression of 30R-GFP (*Figure 4b*). Importantly, co-expression of Zfp106 did not result in a reduction in the expression of 30R-GFP mRNA *Figure 4—figure supplement 1b*), indicating that the suppression of the neurotoxic phenotype in Drosophila was not due to dilution of GAL4 protein. Taken together, these data show that Zfp106 is a potent suppressor of neurodegeneration in a *C9orf72* ALS model, and the results from this gain-of-function model support the idea that Zfp106 functionally interacts with *C9orf72* GGGGCC RNA repeats.

The ability of Zfp106 to suppress the neurotoxic gain-of-function effect of rGGGGCC repeats has important implications for understanding and possibly ameliorating the pathology of *C9orf72* ALS. Previous studies have suggested roles for Zfp106 in rDNA transcription and in pre-rRNA processing (*Ide and Dejardin, 2015*; *Tafforeau et al., 2013*). Zfp106 was also identified as a polyadenylated mRNA binding protein in an interactome capture study aimed at defining an atlas of mammalian mRNA binding proteins (*Castello et al., 2012*). Interestingly, our interactome data also suggest possible roles for Zfp106 in multiple steps of RNA metabolism and processing and interaction with key RNA binding proteins involved in various forms of neurodegeneration. Precisely how disrupting these processes leads to neurodegeneration and exactly how this relates to *C9orf72* ALS/FTD remains to be determined. A simple model is that sequestration of Zfp106 by the expanded repeats disrupts normal Zfp106 function in patients with *C9orf72* ALS in a sponge-like mechanism (*Cooper-Knock et al., 2014*; *Mizielinska and Isaacs, 2014*). Alternatively, one might speculate that Zfp106 plays other functions important in the biology of *C9orf72*, such as influencing dipeptide repeat protein (DPR) production or affecting nucleolar function or nucleocytoplasmic transport via interactions with Nucleolin or Nup107, which have also been implicated in the pathobiology of *C9orf72* ALS (*Boeynaems et al., 2016*; *Freibaum et al., 2015*; *Haeusler et al., 2014*; *Jovičić et al., 2015*; *Kwon et al., 2014*; *Zhang et al., 2015*). Future studies will address the molecular targets and mechanisms of action of Zfp106 in order to gain further insights into the pathogenesis of *C9orf72* ALS and to identify new possible therapeutic targets for neuroprotection.

## Materials and methods

### Plasmids

A murine *Zfp106* cDNA corresponding to NCBI accession number XM_006499044.2 and a murine *Mbnl1* cDNA corresponding to NCBI accession number NM_020007.4 were cloned by PCR from an E13.5 embryonic heart cDNA library; the amplified cDNAs were subcloned into mammalian expression plasmid pcDNA4/TO (Invitrogen, Carlsbad, CA) modified to include N-terminal 3×FLAG-2×STREP tags. The 3×FLAG-2×STREP-Zfp106 cassette was subsequently subcloned into the *Eco*RI/*Xho*I sites of pUAST (*Brand and Perrimon, 1993*; Flybase ID FBmc0000383) for subsequent generation of transgenic flies, and into the *Age*I/*Spe*I sites of HB9-MCS-IRES-EGFP (Addgene plasmid #16283) for generation of motor neuron-specific *HB9-Zfp106* transgenic mice.

### Mice, flies, and cells

The *Zfp106$^\Delta$* allele was generated by CRISPR-mediated genome editing (*Wang et al., 2013*). 5′-atattgtgccacatcgtgtatgg-3′ and 5′-tttttgagccatacacgatgtgg-3′ sgRNAs, which target exon 1 (ENSMUSE00001311370) of mouse *Zfp106*, were transcribed in vitro using the MEGAshortscript T7 kit (Life Technologies, AM1354) and were purified using the MEGAclear kit (Life Technologies, AM1908). Purified sgRNAs and in vitro-transcribed Cas9 mRNA were co-injected into the cytoplasm of fertilized mouse oocytes using standard transgenic technology as described previously (*De Val et al., 2004*). Two independent F0 founders were independently outcrossed to wild-type mice, and F1 offspring were used for subsequent *Zfp106$^\Delta$* intercrosses.

The mouse strain *Zfp106$^{\text{tm1a(KOMP)Wtsi}}$* (RRID:IMSR_KOMP:CSD26348-1a-Wtsi) was generated from ES cell clone EPD0033_4_C03, obtained from the NCRR-NIH supported KOMP Repository (www.komp.org) and generated by the CSD consortium for the NIH funded Knockout Mouse Project (KOMP). The targeted allele contains a promoterless *LacZ-Neo* cassette, which also contains an En2 splicing acceptor (En2 SA) and polyadenylation signal (pA), flanked by FRT sites; the targeted allele also has loxP sites flanking exon 2 (*Testa et al., 2004*; ENSMUSE00000454975). To generate *Zfp106$^{\text{lacz}\Delta/+}$* mice, male *Zfp106$^{\text{laczflox}/+}$* mice were crossed to female *Mef2c*-AHF-Cre mice (*Verzi et al., 2005*; RRID:MMRRC_030262-UNC). When crossed from the female, *Mef2c*-AHF-Cre functions as a universal deleter line due to Cre expression in the female germline, resulting in Cre-dependent recombination in all cells in all offspring (*Ehlers et al., 2014*).

Transgenic *HB9-3×FLAG-2×STREP-Zfp106-IRES-egfp* (*HB9-Zfp106*) mice expressing Zfp106 in motor neurons under the control of the promoter of the *Mnx1* gene (herein called *HB9*), were generated by standard pronuclear injection of the HB9-3×FLAG-2×STREP-Zfp106-IRES-EGFP plasmid linearized by *Xho*I digestion. Injected embryos were implanted into pseudo-pregnant females, and F0 founders were screened by PCR. Grip strength was measured by Kondziella inverted screen test (*Deacon, 2013*; *Kondziella, 1964*).

Genotyping of mutant and transgenic mice was performed by PCR on genomic DNA isolated from tail biopsies using the following primers: *Zfp106$^\Delta$*, 5′-caggaggctgtgctgtgata-3′ and 5′-attctaggactgggaggcaga-3′; *Zfp106$^{\text{lacz}\Delta}$* and *Zfp106$^{\text{laczflox}}$* 5′-cccctgaacctgaaacata-3′ and 5′-gaaagtcccagctgcaagac-3′ for the mutant allele and 5′-tgaatcagctcgtaaacggac-3′ and 5′-atgaccgctcaaatcgatcaa-3′ for the wild type allele; *HB9-3×FLAG-2×STREP-Zfp106-IRES-egfp*, 5′-aagttcatctgcaccaccg-3′ and 5′-tccttgaagaagatggtgcg-3′.

UAS-GFP, UAS-3R-GFP and UAS-30R-GFP transgenic *Drosophila melanogaster* lines were a generous gift of Peng Jin (Emory University) and have been described previously (*Xu et al., 2013*). GMR-GAL4 (FBst0008605) and OK371-GAL4 (FBst0026160) lines were obtained from the Bloomington Stock Center. UAS-3×FLAG-2×STREP-Zfp106 (UAS-Zfp106) transgenic *D. melanogaster* were generated by standard *w+* P-element-mediated transgenesis (*Brand and Perrimon, 1993*) on a *w$^{1118}$* background (BestGene, Inc.). Flies were maintained at 23°C, unless otherwise indicated. For western blot detection of the Zfp106 transgene, flies were shifted to 30°C for 3 days prior to removing heads for protein extraction to induce higher expression of GAL4 from GMR-GAL4.

Mouse Neuro-2a (ATCC, CCL-131; RRID:CVCL_0470) and human HEK293T cells (ATCC, CRL-3216; RRID:CVCL_0063) were obtained directly from ATCC and maintained in Dulbecco's modified Eagle medium supplemented with 10% fetal bovine serum (FBS, Gibco, Waltham, MA) and penicillin-streptomycin (Gibco). All cell lines were routinely tested for *Mycoplasma* contamination with the

MycoAlert PLUS Mycoplasma Detection Kit (Lonza, Switzerland, LT07-705) and found to be negative.

## Biotinylated-RNA pull-down of nuclear proteins

Biotinylated-RNA pull-down of nuclear proteins was performed as previously described with modifications (*Marin-Bejar and Huarte, 2015*). 100 pmol of ssRNA oligomer [either r(GGGGCC)$_8$ or r-(AAAACC)$_8$] biotinylated at the 3' end (Integrated DNA Technologies, Inc.) were incubated with 50 µl of pre-washed Nucleic Acid Compatible Streptavidin Magnetic Beads (Pierce Magnetic RNA-Protein Pull-Down kit, cat. n. 20164) in RNA capture buffer (20 mM Tris, pH 7.5; 1 M NaCl; 1 mM EDTA) for 30 min on a rotary shaker. The beads were washed twice in 20 mM Tris at 4°C for 5 min each and then incubated with nuclear extract for 4 hr at 4°C. Approximately $1 \times 10^7$ Neuro-2a cells were detached from cell culture dishes when they were approximately 80% confluent with 10 mM EDTA in PBS, incubated in Harvest Buffer (10 mM HEPES, pH 7.9; 50 mM NaCl; 0.5 M sucrose; 0.1 mM EDTA; 0.5% NP-40) for 10 min on ice. Nuclei were pelleted by centrifugation at $4500 \times g$ for 10 min at 4°C, washed for 5 min in buffer A (10 mM HEPES, pH 7.9; 10 mM KCl; 0.1 mM EDTA; 0.1 mM EGTA), centrifuged again at $4500 \times g$ for 5 min at 4°C and lysed in IP buffer (50 mM Tris, pH7.4; 150 mM NaCl; 1 mM EDTA) plus 0.5% NP-40. Complete protease inhibitor and PhosSTOP phosphatase inhibitor (Roche) were added to all buffers. After incubation with the nuclear extracts, beads were washed 3× in IP buffer plus 0.05% NP-40, and then 1× in IP buffer without NP-40. Co-precipitated proteins were eluted from bound biotinylated RNAs in 45 µl 50 mM Tris-Cl, pH7.4; 150 mM NaCl; 1 mM EDTA plus 0.05% RapiGest (Waters Corp., Milford, MA). Precipitated proteins were then detected by mass spectrometry or western blot.

## RNA EMSA

100 pmol of r(GGGGCC)$_4$ or r(CUG)$_8$ single-stranded RNA (Integrated DNA Technologies, Inc.) were 5'-end labeled with [γ-$^{32}$P]ATP and T4 Polynucleotide Kinase (New England Biolabs) as previously described (*Rio, 2014*). Unincorporated radioactivity was removed using Illustra MicroSpin G-25 Columns (GE Healthcare Life Sciences, UK). RNA EMSA was then performed as previously described (*Rio, 2014*) by incubating the labeled RNA (100,000 cpm) with purified Zfp106 or MBNL1 protein for 30 min at room temperature in 1× binding buffer (40 mM Tris-Cl, pH 8.0; 30 mM KCl; 1 mM MgCl$_2$; 0.01% NP-40; 1 mM DTT; 5% glycerol; 10 µg/ml bovine serum albumin). The RNA-protein mixtures were then electrophoresed in 6% acrylamide-TBE gels. Gels were dried and exposed to X-ray film for analysis. For the competition experiments, the purified proteins were preincubated for 10 min at room temperature with 30-fold molar excess of the following unlabeled single-stranded RNAs: r(GGGGCC)$_4$, r(AAAACC)$_4$, r(CUG)$_8$, or a mutated (mut) GGGGCC repeat sequence, r(GAGGCCGGGACCGAGACCGAGGCC).

## Histology, immunohistochemistry, and immunofluorescence

Skeletal muscles were isolated from sex- and age-matched mice, mounted on cork discs with 10% tragacanth gum (Sigma) and immediately frozen in liquid nitrogen-cooled isopentane. Frozen muscles were then cryosectioned at a thickness of 10 µm and stained with hematoxylin and eosin as previously described (*Verzi et al., 2005*). For X-gal staining, cryosections were fixed in ice cold 4% formaldehyde, 0.5% glutaraldehyde in PBS for 2 min and then stained as previously described (*Verzi et al., 2005*). Spinal cords were dissected from mice perfused with 4% PFA, additionally post-fixed with 4% PFA overnight, equilibrated with 30% sucrose in PBS and embedded in OCT. Immunostaining for choline acetyltransferase (ChAT) was performed on 40 µm free-floating cryostat sections as described previously (*Qiu et al., 2014*) with goat anti-ChAT antibody (1:300; Millipore, AB144P; RRID:AB_2079751) and biotinylated rabbit anti-goat (1:200; Vector Laboratories, BA-5000; RRID:AB_2336126) as secondary antibody, followed by diaminobenzidine immunoperoxidase development using the Vectastain Elite ABC kit (Vector Laboratories, PK-6105) and the DAB substrate kit for peroxidase (Vector Laboratories, SK-4100).

Post-mortem human spinal cord tissue was obtained from the UCSF Neurodegenerative Disease Brain Bank (NDBB). Formalin fixed paraffin-embedded tissue blocks were cut into 10 µm-thick sections. Tissue sections were subjected to antigen retrieval by autoclaving at 120°C for 5 min in 0.1M citrate buffer, pH 6.0. Immunohistochemistry was performed with rabbit anti-Zfp106 (1:500; Bethyl,

A301-526A; RRID:AB_999690). Following immunostaining, digital images were captured using a high resolution digital camera (Nikon DS-U2/L2) mounted on a Nikon Eclipse80i microscope using NIS-Elements (Version 3.22, Nikon) imaging software. Mouse spinal cords were dissected and then fixed in 2% paraformaldehyde overnight at 4°C, embedded in paraffin and then sectioned at 20 µm for subsequent immunofluorescence analyses. Neuro-2a cells were grown on coverslips and fixed with 4% PFA for 10 min. Immunofluorescence was performed as described previously (*Anderson et al., 2004*) with the following antibodies: mouse anti-NeuN (1:500; Millipore, MAB377; RRID:AB_177621), chicken anti-$\beta$-gal (1:500; Aves, BGL1040; RRID:AB_2313507), rabbit anti-Zfp106 (1:100; Bethyl, A301-527A; RRID:AB_999691). Alexa Fluor 594 goat anti-chicken (1:500; Molecular Probes A11042; RRID:AB_142803), Alexa Fluor 488 donkey anti-mouse (1:500; Molecular Probes A21202; RRID:AB_141607), and Alexa Fluor 594 donkey anti-rabbit (1:500; Molecular Probes A21207 RRID:AB_141637) were used as secondary antibodies. The slides were counterstained and mounted with SlowFade Gold antifade reagent with DAPI (Life Technologies, Waltham, MA). Confocal images were acquired with a Leica TCS SPE laser scanning confocal system.

## Coimmunoprecipitation and western blot

For coimmunoprecipitation, Neuro-2a cells were transfected with 7 µg of 3×FLAG-2×STREP-Zfp106 or 3×FLAG-2×STREP parental vector per 55 cm$^2$ plate by using GenJet in vitro DNA Transfection Reagent for Neuro-2A Cells, according to the manufacturer's recommendations (Signagen, Rockville, MD). Cells were lysed in Benzonase Lysis Buffer [20 mM Tris-HCl (pH 7.5), 40 mM NaCl, 2 mM MgCl$_2$, 0.5% NP-40 and Complete EDTA-free protease inhibitor (Roche, Switzerland)] with the addition of 100 U/ml benzonase (Novagen, Germany) to digest nucleic acids. After incubation for 30 min on ice, NaCl concentration was increased to 150 mM and the lysates were further incubated with agitation for 30 min at 4°C. After clearing, the lysates were incubated with pre-washed anti-FLAG (M2)-conjugated magnetic beads (Sigma, St. Louis, MO, M8823) for 4 hr at 4°C. The beads were then washed five times with IP Buffer with 0.05% NP40, before incubation at 70°C for 10 min in 2× NuPAGE Sample Buffer (ThermoFisher Scientific, Waltham, MA, NP0007) supplemented with NuPAGE Sample Reducing Agent (ThermoFisher Scientific, NP0004) to release coimmunprecipitated proteins. Samples were then subjected to SDS-PAGE and western blot using standard procedures.

For western blot to detect Zfp106 in transgenic *D. melanogaster*, heads from 40 *Gmr-Gal4, UAS-3×FLAG-2×STREP-Zfp106*, and *Gmr-Gal4;UAS-3×FLAG-2×STREP-Zfp106* flies were removed and homogenized in RIPA buffer (50 mM Tris-Cl, pH 7.4; 150 mM NaCl; 2 mM EDTA; 1% NP-40; 0.2% SDS) with a Dounce homogenizer (Wheaton, Millville, NJ). Homogenates were cleared by centrifugation, and total protein content was quantified with the Bio-Rad Protein Assay kit (Bio-Rad, 5000002). An equivalent amount of total protein for each sample was subjected to SDS-PAGE and western blot using standard procedures. The following antibodies were used for western blot: rabbit anti-Zfp106 (1:1000; Bethyl, A301-527A; RRID:AB_999691); mouse anti-FLAG (1:1,000; Sigma, F3165 (clone M2); RRID:AB_439685), mouse anti-actin (1:500; Developmental Studies Hybridoma Bank, JLA20; RRID:AB_528068), rabbit anti-TDP-43 (1:1000; Proteintech, 12892–1-AP; RRID:AB_2200505) and rabbit anti-Nup107 (1:1000; Proteintech, 19217–1-AP; RRID:AB_10597702).

## RT-qPCR

For RT-qPCR, total RNA was extracted from lumbar spinal cord from mice or from 20 flies per genotype using Trizol (Invitrogen). RNA was treated for 1 hr at 37°C with DNaseI, followed by cDNA synthesis using the Omniscript RT kit (Qiagen). qPCR was performed using the MAXIMA SYBR Green kit (Thermo Scientific) and a 7900HT Fast Real Time PCR System (Applied Biosystems). The SDS 2.4 software package was used to extract raw qPCR data. Data were normalized to housekeeping genes by the $2^{-\Delta\Delta Ct}$ method. The following primers were used for qPCR in mice: primers 5′-atttggcgttgtggatcact-3′ and 5′-gcgttcaatatgtcctgcaa-3′ for amplification of *Zfp106* exon 21–22 and 5′-accacagtccatgccatcac-3′ and 5′-tccaccaccctgttgctgta-3′ for amplification of *gapdh*. The following primers were used for qPCR in flies: 5′-gcgcggttactctttcacca-3′ and 5′- atgtcacggacgatttcacg-3′ for *actin*, and 5′-gcacgacttcttcaagtccgccatgcc-3′ and 5′-gcggatcttgaagttcaccttgatgcc-3′ for *egfp*.

## Protein purification, mass spectrometry, and analysis of Zfp106-interacting proteins

HEK293T cells were transfected with 20 µg of 3×FLAG-2×STREP-Zfp106 or 5 µg of 3×FLAG-2×STREP-MBNL1 plasmid per 150 cm$^2$ plate by using PolyJet DNA In Vitro Transfection Reagent, according to the manufacturer's recommendations (Signagen). For RNA EMSA, Zfp106 and MBNL1 were purified from HEK293T total cellular lysates using the Strep-Tag system (*Schmidt and Skerra, 2007*), with Strep-Tactin Sepharose (IBA, catalog #2-1201-010), according to the manufacturer's protocol. 0.1% Triton X-100 was added to all buffers for cell lysis and chromatography to enrich for purified proteins. Eluted fractions were analyzed by SDS-PAGE stained with Coomassie brilliant blue. Comparable amounts of Zfp106 and MBNL1 were used in RNA EMSA. A parallel purification from HEK293T cells transfected with the 3×FLAG-2×STREP parental vector was performed and the corresponding fractions in which Zfp106 and MBNL1 were eluted were used in RNA EMSA as a negative control.

Affinity purification of Zfp106 for identification of interacting proteins by mass spectrometry was performed as previously described (*Jager et al., 2012*). Briefly, cleared nuclear extracts from HEK293T cells were prepared as described above (Biotinylated-RNA pull-down section) with the introduction of a benzonase digestion step to digest nucleic acids (100 U/ml for 30 min in the presence of 1 mM MgCl$_2$) and incubated with pre-washed anti-FLAG(M2)-conjugated magnetic beads (Sigma, M8823) for 2 hr at 4°C. Following purification, complexes bound to beads were washed and then eluted in IP buffer containing 100 µg/ml 3×FLAG peptide (Elim Biopharm) and 0.05% RapiGest (Waters Corp.). To analyze proteins by liquid chromatography-mass spectrometry (LC-MS/MS), the eluates were digested with trypsin, and the peptides were analyzed on a Velos Pro mass spectrometry system (Thermo Scientific). Peptides were matched to protein sequences by the Protein Prospector Algorithm (RRID:SCR_014558), and data were searched against a database containing SwissProt human protein sequences (*UniProt Consortium, 2015*; UniProtKB, RRID:SCR_004426; http://www.uniprot.org). Parallel FLAG-affinity purifications were performed on untransfected HEK293T cells and HEK293T cells transfected with 3xFLAG-2xSTREP tag only, GFP-3xFLAG-2xSTREP or the following 3xFLAG-2xSTREP-tagged proteins: Vif, Tat, MEF2C, Myocardin, and Etv2. The bait-prey datasets were analyzed with the APMS scoring algorithm MiST (*Jager et al., 2012*), and an in-house curated APMS dataset used for determining machine background noise. The MiST algorithm compares peak intensities from the mass spectrum to determine abundance, reproducibility across multiple replicated experiments (n = 5 for each condition), and specificity based on the uniqueness of each interaction, and then these three metrics are used to determine a composite score by the algorithm (*Jager et al., 2012*). The Zfp106 interactome was compiled by selecting bait-prey pairs with a top 5% MiST score. Enrichment analysis for GO biological process terms overrepresented in the interactome was performed on the Gene Ontology Consortium website (*Gene Ontology Consortium, 2015*; RRID:SCR_002811; http://www.geneontology.org). The top 5% MiST hits were visualized as network representation using Cytoscape (*Cline et al., 2007*; RRID:SCR_003032). Protein complex analysis was performed using the manually curated CORUM protein complex database (*Ruepp et al., 2008*; RRID:SCR_002254; http://mips.helmholtz-muenchen.de/corum/).

## Drosophila larval immunofluorescence and quantification of active zones

For analysis of neuromuscular junction (NMJ) phenotypes in Drosophila, wandering third instar larvae were dissected as previously described (*Smith and Taylor, 2011*). Dissected larvae were fixed in 3.7% formaldehyde in PBS for 20 min, washed in PBT (PBS with 0.4% Triton X-100) and blocked with 10% normal goat serum for 1 hr at room temperature. The larvae were then incubated overnight at 4°C with mouse anti-Brp antibody (1:50; Developmental Studies Hybridoma Bank, nc82; RRID:AB_2314865) to label the active zones and Cy3 goat anti-HRP (1:200; Jackson ImmunoResearch, 123-165-021; RRID:AB_2338959) to label the NMJ membrane and axons. Next, samples were washed in PBT and incubated with Alexa Fluor 488 goat anti-mouse (1:200; Molecular Probes, A11001; RRID:AB_2534069) in PBT + 10% normal goat serum at room temperature for 2 hr. Multiple confocal stacks were acquired using a point scanning confocal microscope (Leica TCS SPE) and then processed using the DeadEasy Synapse ImageJ plugin (*Sutcliffe et al., 2013*), which provides total voxel

occupancy by Brp signal at each nerve terminal. Brp signal was quantified at muscle 6/7 NMJs from abdominal segment A3.

## Rapid iterative negative geotaxis (RING) assay

Locomotor activity in 28-day post-eclosion flies was assayed by the Rapid Iterative Negative Geotaxis (RING) assay, as previously described (*Gargano et al., 2005*; *Nichols et al., 2012*). Briefly, 25 flies for each genotype were transferred without anesthetizing in polystyrene vials assembled in a RING apparatus. The RING apparatus was sharply tapped down three times and negative geotaxis was recorded on video for six trials for each genotype. The average height climbed at 3 s after completion of the third tap was scored for each genotype.

## Statistical analyses

Statistical analyses were performed using GraphPad Prism 5.0 (GraphPad Software; RRID:SCR_002798). Data were analyzed by one-way or two-way ANOVA followed by Bonferroni's Multiple Comparison Test or by *t*-test.

## Acknowledgements

We thank Peng Jin, Grae Davis, Biao Wang, and Fen-Biao Gao for reagents, Prasanth Rao, Ajay Chawla, and Rahul Deo for helpful discussions, Jianxin Hu and Reuben Thomas for assistance with sequence analysis and Shan-Mei Xu for technical assistance.

## Additional information

### Funding

| Funder | Grant reference number | Author |
| --- | --- | --- |
| American Heart Association | 14POST1862005 | Barbara Celona |
| Sandler Foundation | UCSF PBBR 7027701 | Barbara Celona |
| National Institutes of Health | HL064658 | Brian L Black |
| National Institutes of Health | HL089707 | Brian L Black |
| Amyotrophic Lateral Sclerosis Association | 17-IIP-358 | Brian L Black |
| National Institutes of Health | P01AG019724 | Bruce L Miller<br>William W Seeley |
| National Institutes of Health | P50AG023501 | Bruce L Miller<br>William W Seeley |

The funders had no role in study design, data collection and interpretation, or the decision to submit the work for publication.

### Author contributions

BC, Conceptualization, Formal analysis, Validation, Funding acquisition, Investigation, Writing—original draft, Writing—review and editing, Final approval of the manuscript; JvD, Formal analysis, Investigation, Writing—original draft, Writing—review and editing, Final approval of the manuscript; SCV, Conceptualization, Formal analysis, Investigation, Writing—review and editing, Final approval of the manuscript; RK, JRJ, AAT, Formal analysis, Investigation, Writing—review and editing, Final approval of the manuscript; AH, Conceptualization, Supervision, Writing—review and editing, Final approval of the manuscript; BLM, Conceptualization, Funding acquisition, Writing—review and editing, Final approval of the manuscript; EJH, Conceptualization, Formal analysis, Supervision, Writing—review and editing, Final approval of the manuscript; NJK, Conceptualization, Formal analysis, Supervision, Methodology, Writing—original draft, Final approval of the manuscript; WWS, Conceptualization, Funding acquisition, Investigation, Writing—review and editing, Final approval of the manuscript; BLB, Conceptualization, Formal analysis, Supervision, Funding acquisition, Investigation, Writing—original draft, Project administration, Writing—review and editing, Final approval of the manuscript

**Author ORCIDs**
Brian L Black, http://orcid.org/0000-0002-6664-8913

**Ethics**
Animal experimentation: All experiments using vertebrate animals were reviewed and approved by the University of California, San Francisco Institutional Animal Care and Use Committee (IACUC) under protocols AN108111 and AN087046, and all animal research complied with all institutional and federal guidelines.

## Additional files

**Supplementary files**
• Supplementary file 1. Zfp106 interactors identified in HEK293T cells. 39 proteins (indicated by Uniprot accession and gene name) identified with 95% confidence based on MiST score by affinity-purification mass spectrometry of FLAG-Zfp106 in HEK293T cells.

• Supplementary file 2. Protein complexes overrepresented in the Zfp106 interactome. List of protein-protein pairs in protein complexes from the CORUM database that are overrepresented in the Zfp106 interactome. Proteins are indicated by Uniprot accession number and gene name, protein complexes by CORUM protein complex ID and name.

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
