## [Decision Letter]

[Editors’ note: this article was originally rejected after discussions between the reviewers, but the authors were invited to resubmit after an appeal against the decision.]

Thank you for submitting your work entitled "Suppression of *C9orf72* RNA repeat-induced neurotoxicity by the ALS-associated zinc finger protein Zfp106" for consideration by *eLife*. Your article has been favorably evaluated by Marianne Bronner as the Senior Editor and three reviewers, one of whom is a member of our Board of Reviewing Editors. The reviewers have opted to remain anonymous.

Our decision has been reached after consultation between the reviewers. Based on these discussions and the individual reviews below, we regret to inform you that your work will not be considered further for publication in *eLife*.

*Reviewer #1:*

In this concise and well written paper, Celona et al. present extensive evidence that the zinc finger protein Zfp106 is a major player in *C9orf72* RNA repeat-induced neurotoxicity (expanded GGGGCC repeats in the first intron of the *C9orf72* gene represent the most common cause of familial amyotrophic lateral sclerosis (ALS)). Starting with a biotinylated RNA pulldown of neural nuclear proteins, the authors identify Zfp106. They go on to show that Zfp106 loss-of-function causes an ALS-like phenotype in mice, and that it can be suppressed by transgenic restoration of Zfp106 specifically in motor neurons. They also show that Zfp106 can suppress neurotoxicity in a fly model of *C9orf72* ALS, and that Zfp106 interacts with multiple other RNA binding proteins including the ALS factors TDP-43 and FUS. The data appear convincing and highly significant.

*Reviewer #2:*

*C9orf72* has attracted a lot of attention recently because of its strong association with familial ALS. Two major models by which the repeats can cause disease have already been proposed, and this manuscript provides some new evidence for one of those models – the RNA binding protein sequestration model.

This manuscript opens with the observation that the zinc finger protein Zfp106 binds the repeats in *C9orf72* in an in vitro pull down assay. Loss of Zfp106 was already known from knockout mice to cause a nondevelopmental neuromuscular degeneration, but on the basis of this biochemical interaction the authors now propose that this is an ALS-like degeneration caused by motor neuron degeneration. They show evidence for loss of motor neurons in the knockout mice and consistent with Zfp106 functioning in motor neurons, they show some phenotypic rescue of weight and grip strength with rescue of Zfp106 expression in motor neurons. In order to link Zfp106 more specifically to an ALS model of neuromuscular disease they employ a well-established *Drosophila* model of *C9orf72* repeat overexpression in motor neurons and show phenotypic rescue with Zfp106 overexpression. Finally, to suggest a mechanism for Zfp106 function they do mass spec of Zfp106 co-associated proteins in heterologous HEK293 cells and find lots of RNA binding and processing factors.

Although Zfp106 knockout mice had already been extensively studied for their neuromuscular phenotype, the link revealed in this study between Zfp106 and *C9orf72* is novel and adds a key new dimension to that observation. However that being said, the investigation of this association in this manuscript is somewhat superficial weakening the impact of the finding.

1) The RNA assays in Figure 1 are convincing that Zfp106 can bind the repeats but with just a single control shown it is not clear how specific this interaction might be. Do other RNA binding proteins also bind these repeats in this assay? Can Zfp106 be shown not to bind some other RNA sequence that is the target of a different RNA binding protein?

2) The effect of Zpf106 loss on motor neuron degeneration is insufficiently characterized. The number of ChAT neurons in Figure 2 should be quantified across a set of mice, and histological validation of their degeneration should be shown. In addition, there is no data in the paper to show that the phenotypic rescue with the H89 transgene does actually result in rescue of this motor neuron phenotype and that must be shown.

3) The mass spec experiments in Figure 3 add little to the model because they fail to test whether the interactions seen are specific to Zfp106. The control is simply the empty vector, whereas a more meaningful control would be another RNA binding protein that does not show the same specificity. The authors do not validate any of the proposed interactions discovered by mass spec using pulldowns and immunoblotting, which would help to determine whether they are direct or perhaps indirect and mediated by multiple proteins binding a single RNA.

4) Although the fly model has already been established as motor neuron specific, the authors should still show that when they rescue the phenotypes with Zfp106 overexpression that this results in improved motor neuron survival.

*Reviewer #3:*

Celona et al. reported the identification of Zfp106, a Zinc-finger-containing protein, as an interactor of G4C2 HRE. Loss of Zfp causes motor deficits and neurodegeneration that mimics ALS in mice that can be rescued by restoring the expression in motor neurons. Protein interactome analysis suggested that Zfp is associated with RNA binding proteins that has been implicated in ALS. In addition, Zfp overexpression rescues G4C2-mediated motor neuron toxicity in a fly model of C9-ALS. The finding is important, the story is clean, and the interpretation is clear.

However – one major caveat – none of the mouse studies are novel as the same mouse was recently published: http://www.pnas.org/content/early/2016/07/13/1608423113.long. The authors will need cite this new preceding mouse work and provide information on what new data their study offers (and detail the lessor contribution of the fly studies).

We have several suggestions to improve the manuscript.

1) Please label the age of mice on the sections/stainings.

2) Please briefly introduce the method used to express Zfp in motor neurons in the main text, which Cre line used etc.

3) Please verify some of the interactions between Zfp and RNA binding proteins, e.g. TDP-43 and FUS. Please stain and blot TDP-43 and FUS in Zfp lof mice to show whether they are mislocalized and/or changed in proteins levels.

4) The real relevance to human disease would be more convincing by an examination of far more than just two human cases. The author must show reproducibility of this pathology by and examination of at least 8-15 C9 cases to be sure of the rigor of the observation and need to include disease control such as: sporadic ALS, and even mutant SOD1 ALS spiral cord. Is this just an exceptionally rare neuropathological observation – how common is it? Is it seen in glia? Which glia? Is it found in unaffected brain regions? Finally, they need to provide more details the human cases (e.g. age, length of disease, sex, post mortem interval, etc.).

5) Please show G4C2 mRNA is unchanged when co-expressing Zfp in G4C2 flies to make sure the rescue is not due to a GAL4-UAS dilution effect. Also, fly doesn't have a gene named Zfp106, please indicate which gene (CG number) is referred to. If they are referring to the mouse/human Zfp106, please specify which is the fly homolog and the authors need to express the fly homolog to show the same rescue. Otherwise, it's unknown what a foreign protein is doing in fly. Once this homolog is identified, please express its RNAi in G4C2 flies to see if it enhances phenotypes.

[Editors’ note: what now follows is the decision letter after the authors submitted for further consideration.]

Thank you for resubmitting your work entitled "Suppression of *C9orf72* RNA repeat-induced neurotoxicity by the ALS-associated RNA-binding protein Zfp106" for further consideration at *eLife*. Your revised article has been favorably evaluated by Marianne Bronner as the Senior Editor, a Reviewing Editor, and two reviewers.

The manuscript has been improved but there are some remaining issues, and one serious reservation (#7), that need to be addressed before a final decision may be reached.

Remaining revisions needed:

In this revision the authors have added significant new data to strengthen their story. However, there are still parts of the manuscript that need further textual modifications for the sake of clarification, and the revision of the mass spec section of the paper did not fully resolve the concerns I had raised about the interpretation of these data. Further, as stated in #7 below, you may need further experimental evidence to support your claim of functionally related fly and mouse Zfp othologs.

1) The text frequently overstates the relationship between the model organism phenotypes and ALS. For example in the Abstract it says "Zpf106 knockout mice develop ALS". The mice develop motor neuron degeneration and an ALS-like motor phenotype, but it is going too far to call it ALS. There are numerous instances of this usage throughout the text, not enumerated here, and all should be corrected.

2) With respect to the molecular validation of the CRISPR mutant mouse, I appreciate that PCR has been added to Figure 2—figure supplement 3. However, it remains unclear from the descriptions provided in the paper exactly what mutations were made to the Zpf106 gene and what this figure really shows. The description of the CRISPR mutation suggests this was two point mutations in exon 1 generated in separate lines, that were then crossed to each other to make a KO? It sounds from the methods as if the primers were against exons 21-22 far away from the mutation itself. If this is a point mutation mouse then why would the RNA be reduced? And if these are both null lines, why is the RNA expression for both the KO and the CRISPR mouse not completely gone (since these are not cell type specific)? I'm guessing that the answer to both of these questions is nonsense-mediated decay, but I thought that was uncommon with the CRISPR point mutations and anyway if this is the case then better to explain in the text than to have the reader fill in the gaps. Finally, given that the authors have an antibody against Zpf106 (Figure 1) the authors should show loss of protein to validate that these actually are null alleles.

3) In Figure 2 and the associated supplementary figures it is not always clear which Zpf106 strain is being used. Is the "knockout" the CRISPR mouse, and only the cases where the strain is annotated for "lacZ" the KOMP mouse? This naming scheme should be more explicitly spelled out at least in the Methods if not in the Results text since they are both presumably "knockouts" or "nulls".

4) With respect to m the mass spec experiments, it is now mentioned in the sixth paragraph of the Results and Discussion section that "several controls were included in the mass spectrometry experiments, including the DNA-binding factors Etv2 and MEF2C and the RNA-binding proteins Vif and Tat", and the Methods section describes that these proteins were expressed and pulled down. Yet the manuscript never says how these were used as controls. Were these negative controls meaning that the authors subtracted from their list of pulldown proteins anything that came down with any other protein? Were they specificity controls meaning that a comparative analysis of binding proteins was carried out? Were they positive controls, meaning the authors knew what proteins were supposed to interact with these and found them in the pulldown? It is not enough to say "controls were run" and then never show the data or never explain how those data were used.

5) The direct IP experiments provide important support for the results of the mass spec. However, this is exactly where it would be helpful to see some of those positive and negative controls used.

6) The authors should not call their Zfp mice "ALS" in the Abstract as they're simply describing a motor neuron-like disease. They did not show in their mice upper motor neuron involvement (by definition required to call a disease ALS) and their muscle pathology is not seen in typical ALS. To compensate for it, they should examine human tissue to see if real C9 ALS has changes in Zfp106- otherwise this is simply a mouse model of interesting biology – but no definite link to true human disease. Because of the failure of mouse models to truly predict human disease- or therapies- a direct comparison to actual human ALS tissue is required for all studies if trying to make such association.

7) The authors did not answer the questions whether there is a fly orthologue of Zfp and, if yes, what is its CG number. (This is not trivial, as FlyBase curators may ask you even after you publish your paper.) If there is one such orthologue and if this orthologue does not rescue neurodegeneration in C9 flies, then the rescue effect of mouse Zfp showed by the authors is not through Zfp evolutionarily conserved functions, but possibly an artifact, which raises a critical concern about the study's translatability. Without any evidence that the mouse Zfp is functional in flies (e.g. is it properly localized, post-translationally modified, or regulating the same genes as it normally does in mice?), the data that this protein rescues neuronal defects in C9-ALS flies may not be biologically meaningful at all. The authors in this case need to test whether mouse Zfp rescues neurodegeneration in C9-ALS mice. Otherwise, they need to identify downstream genes through which the mouse Zfp suppresses C9 defects in flies and then validate in mammals that the orthologues of these genes are indeed downstream targets of Zfp. In addition, the rebuttal is confusing because it claims that the authors did not argue a "rescue", but rather a "suppression" of C9-phenotypes (although I did not see a profound difference between them), whereas in the legend of Figure 4, the authors said "this defect was completed rescued by co-expression of Zfp106".

---

## [Author Response]

[Editors’ note: the author responses to the first round of peer review follow.]

*Reviewer #2: […] Although Zfp106 knockout mice had already been extensively studied for their neuromuscular phenotype, the link revealed in this study between Zfp106 and C9orf72 is novel and adds a key new dimension to that observation. However that being said, the investigation of this association in this manuscript is somewhat superficial weakening the impact of the finding.*

*1) The RNA assays in Figure 1 are convincing that Zfp106 can bind the repeats but with just a single control shown it is not clear how specific this interaction might be. Do other RNA binding proteins also bind these repeats in this assay? Can Zfp106 be shown not to bind some other RNA sequence that is the target of a different RNA binding protein?*

These are reasonable suggestions for additional controls related to the specificity of Zfp106-RNA interaction. Accordingly, we performed two additional EMSA experiments. The first of these new EMSA experiments shows that Zfp106 binding to GGGGCC repeats is efficiently competed by unlabeled self probe but not by a mutant RNA probes or by CUG repeats (muscleblind/MBLN1 binding sequence) even when those unlabeled control probes are added at 30-fold molar excess. That experiment (new Figure 1—figure supplement 1) also shows that MBLN1 protein does not bind to GGGGCC repeats under the same conditions in which Zfp106 efficiently binds. In a complementary experiment (new Figure 1—figure supplement 1), a second new EMSA shows clearly that MBLN1 binds specifically and efficiently to CUG repeats and that under those same conditions, Zfp106 shows no discernable binding.

*2) The effect of Zpf106 loss on motor neuron degeneration is insufficiently characterized. The number of ChAT neurons in Figure 2 should be quantified across a set of mice, and histological validation of their degeneration should be shown. In addition, there is no data in the paper to show that the phenotypic rescue with the H89 transgene does actually result in rescue of this motor neuron phenotype and that must be shown.*

These are good suggestions. Accordingly, we have quantified the number of ChAT+ motor neurons from wild type and *Zfp106* knockout mice. These new data, which are shown in Figure 2 in the revised manuscript, demonstrate a statistically significant reduction in ChAT+ motor neurons in *Zfp106* knockout mice. We also examined motor neuron survival by ChAT staining in *HB9-Zfp106* transgenic rescue mice compared to wild type and *Zfp106* knockout mice without the transgene, and we quantified the number of ChAT+ motor neurons in mice of each of these genotypes. These new data, which show significant rescue of ChAT+ motor neurons upon restoration of Zfp106, are shown in Figure 2 and Figure 2—figure supplement 4 of the revised manuscript.

*3) The mass spec experiments in Figure 3 add little to the model because they fail to test whether the interactions seen are specific to Zfp106. The control is simply the empty vector, whereas a more meaningful control would be another RNA binding protein that does not show the same specificity. The authors do not validate any of the proposed interactions discovered by mass spec using pulldowns and immunoblotting, which would help to determine whether they are direct or perhaps indirect and mediated by multiple proteins binding a single RNA.*

We appreciate the reviewer’s concern regarding the specificity of the mass spectrometry experiments and the need to analyze other RNA binding proteins as a measure of specificity. Indeed, this was done in the original study, although we did not draw enough attention to this key point in the original version of the paper. In the Materials and methods section of the original version of the paper, we indicated that we used several negative controls in the mass spectrometry experiments, including DNA-binding factors Etv2 and MEF2C and the RNA-binding proteins Vif and tat. Importantly, the MIST algorithm, which was used to determine the specificity of the Zfp106 protein-protein interactions, considered the Vif and Tat interactomes, essentially subtracting non-specific and background interactors with those two proteins from the Zfp106 interactome. In the revised manuscript, we have added an additional statement to the Results and Discussion section (sixth paragraph) to draw additional attention to that fact. In addition, as suggested by the reviewer (and reviewer 3), we validated the interaction of Zfp106 with the ALS-associated RNA-binding proteins TDP-43 and Nup107, as examples of interacting proteins, by co-immunoprecipitation. These new data are now shown as Figure 3—figure supplement 1 and are discussed in the seventh paragraph of the Results and Discussion section.

*4) Although the fly model has already been established as motor neuron specific, the authors should still show that when they rescue the phenotypes with Zfp106 overexpression that this results in improved motor neuron survival.*

This is a good suggestion to provide additional depth to these studies. In the revised manuscript, we show that expression of mammalian Zfp106 in the fly suppresses the loss of motor neurons caused by expression of 30 copies of *C9orf72* GGGGCC repeats. Specifically, using established approaches (Smith and Taylor, 2011; Zhang et al., 2015), we now show that 30x GGGGCC expression reduces active zones as measured by anti-Brp staining and importantly that co-expression of Zfp106 restores Brp expression at active zones. These new data are shown as Figure 4 in the revised manuscript.

*Reviewer #3:*

*Celona et al. reported the identification of Zfp106, a Zinc-finger-containing protein, as an interactor of G4C2 HRE. Loss of Zfp causes motor deficits and neurodegeneration that mimics ALS in mice that can be rescued by restoring the expression in motor neurons. Protein interactome analysis suggested that Zfp is associated with RNA binding proteins that has been implicated in ALS. In addition, Zfp overexpression rescues G4C2-mediated motor neuron toxicity in a fly model of C9-ALS. The finding is important, the story is clean, and the interpretation is clear.*

*However – one major caveat – none of the mouse studies are novel as the same mouse was recently published: http://www.pnas.org/content/early/2016/07/13/1608423113.long. The authors will need cite this new preceding mouse work and provide information on what new data their study offers (and detail the lessor contribution of the fly studies).*

We greatly appreciate the reviewer’s recognition of the significance of our work and the positive comments about the findings and interpretations. With regard to the major caveat that “none of the mouse studies are novel,” please note that the very first appearance of the Advance Online version of the cited Olson *PNAS* manuscript was on July 14, 2016. Our *eLife* submission date was June 17 – nearly a month earlier. From a discussion with Dr. Olson at a recent meeting, we were aware of his pending manuscript and specifically chose to submit to *eLife* based on *eLife’s* editorial policy that submitted papers cannot be “scooped” by articles published after review at *eLife* has commenced.

Thus, I do not believe that this single major caveat should be held against our manuscript as a valid reason for rejection. Moreover, the criticism is not quite accurate. In our manuscript, unlike the Olson paper, we show transgenic rescue of the mouse phenotype via restoration of Zfp106 specifically to motor neurons. This is an important and novel finding not made in the other manuscript. Additionally, unlike in the other work, we identify the Zfp106 protein-protein interactome and, very importantly, we show suppression of *C9orf72* repeat-mediated neurotoxicity by Zfp106 in an established *Drosophila* model of *C9orf72* ALS. We disagree that this work in the fly is a lessor contribution.

Of course, it would have been impossible for us to cite the Olson manuscript in the original manuscript since our paper was submitted nearly a month prior to any appearance of the Olson paper, even online. However, as requested by the reviewer, in the revised manuscript, we have cited the Olson PNAS paper and its contribution to our understanding of *Zfp106* genetic function in mice. In addition, we have addressed all of the reviewer’s other concerns, as described below.

*We have several suggestions to improve the manuscript.*

*1) Please label the age of mice on the sections/stainings.*

We have labeled the ages of the mice on all sections in the revised manuscript.

*2) Please briefly introduce the method used to express Zfp in motor neurons in the main text, which Cre line used etc.*

Although this was done in the Methods section in the original version of the manuscript, we have taken the reviewer’s suggestion and added these details in brief to the Results and Discussion section. To clarify, this was a transgenic rescue, in which Zfp106 expression was under direct control of the motor neuron-restricted *HB9* promoter; no Cre lines were used for this rescue.

*3) Please verify some of the interactions between Zfp and RNA binding proteins, e.g. TDP-43 and FUS. Please stain and blot TDP-43 and FUS in Zfp lof mice to show whether they are mislocalized and/or changed in proteins levels.*

As suggested by the reviewer (and by reviewer 2), we validated the interaction of Zfp106 with two ALS-associated RNA-binding proteins, TDP-43 and Nup107, by co-immunoprecipitation. These new data are now shown as Figure 3—figure supplement 1 and are discussed in the sixth paragraph of the Results and Discussion section. We believe that further histological analyses of TDP-4, Fus, and other Zfp106-interacting protein localization in knockout mice deal primarily with details of the ALS phenotype and are beyond the focus of this Short Report, which highlights the genetic function of Zfp106, it’s protein-protein interactome, and the key finding that gain-of-function expression of mammalian Zfp106 was sufficient to suppress neurotoxicity in an established *Drosophila* model of *C9orf72* ALS.

4) The real relevance to human disease would be more convincing by an examination of far more than just two human cases. The author must show reproducibility of this pathology by and examination of at least 8-15 C9 cases to be sure of the rigor of the observation and need to include disease control such as: sporadic ALS, and even mutant SOD1 ALS spiral cord. Is this just an exceptionally rare neuropathological observation – how common is it? Is it seen in glia? Which glia? Is it found in unaffected brain regions? Finally, they need to provide more details the human cases (e.g. age, length of disease, sex, post mortem interval, etc.).

I believe the reviewer may have misinterpreted the data presented in Figure 1’ – no “cases” are shown in our manuscript. In that figure, we present images that show expression of ZNF106 in human tissue, and the two major patterns of subcellular distribution are observed. Based on the reviewer’s comment, I believe that the reviewer may have misinterpreted those data to be *C9orf72* and control patient samples showing pathology. However, only control/non-ALS human post-mortem tissues are shown. The point was not to show pathology but to examine the pattern of expression of the human ortholog. Importantly, the pattern of expression is consistent with a possible role in ALS, but an examination of *C9orf72* patient pathology is beyond the scope of this Short Report.

*5) Please show G4C2 mRNA is unchanged when co-expressing Zfp in G4C2 flies to make sure the rescue is not due to a GAL4-UAS dilution effect. Also, fly doesn't have a gene named Zfp106, please indicate which gene (CG number) is referred to. If they are referring to the mouse/human Zfp106, please specify which is the fly homolog and the authors need to express the fly homolog to show the same rescue. Otherwise, it's unknown what a foreign protein is doing in fly. Once this homolog is identified, please express its RNAi in G4C2 flies to see if it enhances phenotypes.*

The reviewer’s concern about the somewhat remote possibility that co-expression of Zfp106 in the same cells as 30x GGGGCC could cause a GAL4 dilution effect is a valid one. Accordingly, we performed qPCR analysis of UAS-transgene-encoded *gfp* expression in the presence and absence of the additional UAS-*Zfp106* transgene. These new data clearly show the same level of 30R-GFP mRNA in progeny with Zfp106 co-expression as in those without Zfp106 co-expression. These important new control data are included in the revised manuscript as Figure 4—figure supplement 1and are discussed in the eighth paragraph of the Results and Discussion section. With regard to the reviewer’s concern about whether the Zfp106 expression experiments in the fly were using the fly ortholog (if one even exists) or the mouse cDNA, we have clarified in the revised manuscript that indeed this was an experiment using the mouse cDNA. We disagree strongly with the reviewer’s belief that expression of a fly ortholog is essential for the interpretation of these experiments – the expression of mouse Zfp106 in the fly is not a *rescue* of the fly ortholog (if one even exists) and was not purported to be a rescue in our manuscript. This experiment was used solely to examine the interaction of mammalian Zfp106 with the C9orf72 repeats in an established neurodegeneration model in the fly. By design, this was used in much the same way that gain-of-function experiments in cell culture are used. Importantly, in this well-established model, we show profound *suppression* of the *C9orf72* repeat-induced phenotype. This is a key result, which we believe stands on its own as presented.

[Editors’ note: the author responses to the re-review follow.]

*The manuscript has been improved but there are some remaining issues, and one serious reservation (#7), that need to be addressed before a final decision may be reached. Given that the manuscript has already been revised twice, we must insist that this represents the last opportunity you will have to address the remaining concerns.*

Remaining revisions needed:

*In this revision the authors have added significant new data to strengthen their story. However, there are still parts of the manuscript that need further textual modifications for the sake of clarification, and the revision of the mass spec section of the paper did not fully resolve the concerns I had raised about the interpretation of these data. Further, as stated in #7 below, you may need further experimental evidence to support your claim of functionally related fly and mouse Zfp othologs.*

*1) The text frequently overstates the relationship between the model organism phenotypes and ALS. For example in the Abstract it says "Zpf106 knockout mice develop ALS". The mice develop motor neuron degeneration and an ALS-like motor phenotype, but it is going too far to call it ALS. There are numerous instances of this usage throughout the text, not enumerated here, and all should be corrected.*

We edited the description of the Zfp106 knockout phenotype exactly as suggested by the reviewer. We no longer refer to it as ALS in the Abstract or elsewhere in the manuscript. Instead, we refer to the phenotype more generally as neurodegeneration or neurodegeneration and an ALS-like motor phenotype.

*2) With respect to the molecular validation of the CRISPR mutant mouse, I appreciate that PCR has been added to Figure 2—figure supplement 3. However, it remains unclear from the descriptions provided in the paper exactly what mutations were made to the Zpf106 gene and what this figure really shows. The description of the CRISPR mutation suggests this was two point mutations in exon 1 generated in separate lines, that were then crossed to each other to make a KO? It sounds from the methods as if the primers were against exons 21-22 far away from the mutation itself. If this is a point mutation mouse then why would the RNA be reduced? And if these are both null lines, why is the RNA expression for both the KO and the CRISPR mouse not completely gone (since these are not cell type specific)? I'm guessing that the answer to both of these questions is nonsense-mediated decay, but I thought that was uncommon with the CRISPR point mutations and anyway if this is the case then better to explain in the text than to have the reader fill in the gaps. Finally, given that the authors have an antibody against Zpf106 (Figure 1) the authors should show loss of protein to validate that these actually are null alleles.*

We apologize for the confusion. As suggested by the reviewer, we have clarified exactly which Zfp106 knockout allele was used in the manuscript. Simply put, all data shown in the paper use the KOMP allele, and this is now clearly indicated in the third paragraph of the Results and Discussion section. The CRISPR- generated allele was used as an independent validation, exhibiting the same phenotype but not shown in the manuscript. This is now clearly and plainly discussed in the fourth paragraph of the Results and Discussion section. This does not affect any of the data in the manuscript nor does it affect any of the manuscript’s conclusions. Our CRISPR strategy produced frameshift deletion mutations in exon1 causing premature stop codons, likely resulting in nonsense-mediated decay. Our qPCR data show that Zfp106 mRNA is significantly and profoundly reduced in knockout animals; adding additional validation by western blot adds little to the manuscript and will not affect the conclusions of the manuscript or the validity of the experiment. The mice show a profound phenotype, and the mRNA is drastically reduced in knockout animals. Additionally, the ~80% reduction in *Zfp106* expression is consistent with previous studies (Anderson et al., 2016; Joyce et al., 2016); the ~20% residual expression in *Zfp106*-null tissue is likely due to transcripts generated from an alternative promoter and TSS described for the *Zfp106* gene (Grasberger and Bell, 2005).

*3) In Figure 2 and the associated supplementary figures it is not always clear which Zpf106 strain is being used. Is the "knockout" the CRISPR mouse, and only the cases where the strain is annotated for "lacZ" the KOMP mouse? This naming scheme should be more explicitly spelled out at least in the Methods if not in the Results text since they are both presumably "knockouts" or "nulls".*

As noted above (response to point 2), all *Zfp106* knockout data shown in the manuscript were obtained with the KOMP allele. This is now clearly stated in the third paragraph of the Results and Discussion section.

*4) With respect to m the mass spec experiments, it is now mentioned in the sixth paragraph of the Results and Discussion section that "several controls were included in the mass spectrometry experiments, including the DNA-binding factors Etv2 and MEF2C and the RNA-binding proteins Vif and Tat", and the Methods section describes that these proteins were expressed and pulled down. Yet the manuscript never says how these were used as controls. Were these negative controls meaning that the authors subtracted from their list of pulldown proteins anything that came down with any other protein? Were they specificity controls meaning that a comparative analysis of binding proteins was carried out? Were they positive controls, meaning the authors knew what proteins were supposed to interact with these and found them in the pulldown? It is not enough to say "controls were run" and then never show the data or never explain how those data were used.*

The controls used in the mass spectrometry experiments included both DNA and RNA binding proteins and were essentially negative controls relative to the Zfp106 interactome, but the statistical algorithm used is somewhat more nuanced than simply defining controls as negative or positive. The MiST algorithm (Jager et al., 2012) compares peak intensities from the mass spectrum to determine abundance, reproducibility across multiple replicated experiments, and specificity based on the uniqueness of each interaction, and then these three metrics are used to determine a composite score by the algorithm. This is a widely used and well-validated approach for evaluating mass spectrometry data with more than 30 publications citing the effectiveness of the MiST algorithm. We have cited the original publication in the revised manuscript where the details of MiST are discussed. We have also added additional detail the revised manuscript in the Results and Discussion section (sixth paragraph) and to the Methods section (subsection “Protein purification, mass spectrometry, and analysis of Zfp106-interacting proteins”, last paragraph) to clarify how the controls were used.

*5) The direct IP experiments provide important support for the results of the mass spec. However, this is exactly where it would be helpful to see some of those positive and negative controls used.*

We appreciate the reviewer’s recognition that the co-immunoprecipitation experiments, added in response to the prior reviews, provide important support for the mass spectrometry results. These new co-IP experiments include negative control immunoprecipitations using lysates from cells transfected with 3×FLAG-2×STREP tag only, showing the specificity of the co-IP of Nup107 and TDP-43 by 3×FLAG-2×STREP-Zfp106.

*6) The authors should not call their Zfp mice "ALS" in the Abstract as they're simply describing a motor neuron-like disease. They did not show in their mice upper motor neuron involvement (by definition required to call a disease ALS) and their muscle pathology is not seen in typical ALS. To compensate for it, they should examine human tissue to see if real C9 ALS has changes in Zfp106- otherwise this is simply a mouse model of interesting biology – but no definite link to true human disease. Because of the failure of mouse models to truly predict human disease- or therapies- a direct comparison to actual human ALS tissue is required for all studies if trying to make such association.*

As suggested by the reviewer, we no longer refer to the phenotype in *Zfp106* knockout mice as having ALS. We now discuss the phenotype as neurodegeneration or neurodegeneration with an ALS-like motor phenotype. This has been edited throughout the manuscript.

*7) The authors did not answer the questions whether there is a fly orthologue of Zfp and, if yes, what is its CG number. (This is not trivial, as FlyBase curators may ask you even after you publish your paper.) If there is one such orthologue and if this orthologue does not rescue neurodegeneration in C9 flies, then the rescue effect of mouse Zfp showed by the authors is not through Zfp evolutionarily conserved functions, but possibly an artifact, which raises a critical concern about the study's translatability. Without any evidence that the mouse Zfp is functional in flies (e.g. is it properly localized, post-translationally modified, or regulating the same genes as it normally does in mice?) the data that this protein rescues neuronal defects in C9-ALS flies may not be biologically meaningful at all. The authors in this case need to test whether mouse Zfp rescues neurodegeneration in C9-ALS mice. Otherwise, they need to identify downstream genes through which the mouse Zfp suppresses C9 defects in flies and then validate in mammals that the orthologues of these genes are indeed downstream targets of Zfp. In addition, the rebuttal is confusing because it claims that the authors did not argue a "rescue", but rather a "suppression" of C9-phenotypes (although I did not see a profound difference between them), whereas in the legend of Figure 4, the authors said "this defect was completed rescued by co-expression of Zfp106".*

There is no established ortholog of *Zfp106* in *Drosophila* and, despite the reviewer’s assertion, we have never claimed that there is a functionally-related ortholog of Zfp106 in the fly or that we are using Zfp106 to rescue a fly gene. To clarify, we are using *Drosophila* in these experiments solely as a gain-of-function model – there is no clear ortholog of *C9orf72* or the expanded hexanucleotide repeat sequence from *C9orf72* in the fly, yet the overexpression model used here and elsewhere is accepted as a gain-of-function model for *C9orf72*-mediated neurodegeneration/neurotoxicity. Similarly, we use a purely gain-of-function approach with mammalian *Zfp106* expression in the fly to examine whether the interaction of Zfp106 with GGGGCC repeats is a functional interaction. We have further clarified the gain-of-function design of our approach in eighth paragraph of the Results and Discussion section, and have emphasized that Zfp106 suppresses rather than rescues repeat-mediated neurotoxicity.

Generation of multiple new *Drosophila* transgenic lines with a yet to be identified and validated *Drosophila* ortholog or the identification of genes affected by Zfp106 in both flies and mammals is clearly beyond the scope of this Short Report and cannot be completed in a reasonable time frame. Moreover, these extensive new experiments would not change the key conclusion that Zfp106 functionally interacts with the hexanucleotide repeats to suppress neurotoxicity in the fly. This key result, taken together with RNA EMSA and other experiments, which show a biochemical interaction between Zfp106 and the hexanucleotide repeats, clearly establish the physical and functional interaction of Zfp106 with GGGGCC repeats.